# Macrophages as a Therapeutic Target in Metastatic Prostate Cancer: A Way to Overcome Immunotherapy Resistance?

**DOI:** 10.3390/cancers14020440

**Published:** 2022-01-16

**Authors:** Clara Martori, Lidia Sanchez-Moral, Tony Paul, Juan Carlos Pardo, Albert Font, Vicenç Ruiz de Porras, Maria-Rosa Sarrias

**Affiliations:** 1Innate Immunity Group, Germans Trias i Pujol Research Institute (IGTP), Ctra. Can Ruti-Camí de les Escoles s/n, 08916 Badalona, Spain; cmartori@igtp.cat (C.M.); lsanchez@igtp.cat (L.S.-M.); tpaul@igtp.cat (T.P.); 2Department of Medical Oncology, Catalan Institute of Oncology, University Hospital Germans Trias i Pujol, Ctra. Can Ruti-Camí de les Escoles s/n, 08916 Badalona, Spain; jcpardor@iconcologia.net (J.C.P.); afont@iconcologia.net (A.F.); 3Badalona Applied Research Group in Oncology (B·ARGO), Catalan Institute of Oncology, Ctra. Can Ruti-Camí de les Escoles s/n, 08916 Badalona, Spain; 4Germans Trias i Pujol Research Institute (IGTP), Ctra. Can Ruti-Camí de les Escoles s/n, 08916 Badalona, Spain; 5Network for Biomedical Research in Hepatic and Digestive Diseases (CIBERehd), 28029 Madrid, Spain

**Keywords:** metastatic prostate cancer, immunotherapy, immune checkpoint inhibitors, tumor-associated macrophages, tumor microenvironment, immunotherapy resistance

## Abstract

**Simple Summary:**

In recent years, therapeutic options for patients with metastatic prostate cancer have improved significantly. However, the efficacy of current immunotherapy strategies in metastatic prostate cancer patients is limited. The prostate cancer tumor microenvironment, which includes immunosupressive cells such as tumor-associated macrophages, has been proposed as a major barrier to the effectiveness of immunotherapy. Thus, macrophages have emerged as a promising target to directly reduce tumor progression and overcome immunotherapy resistance. In this review we will summarize the current status of therapies targeting macrophages as well as their potential to increase immunotherapy efficacy in metastatic prostate cancer.

**Abstract:**

Prostate cancer (PC) is the most common malignancy and the fifth cause of cancer death in men. The treatment for localized or locally advanced stages offers a high probability of cure. Even though the therapeutic landscape has significantly improved over the last decade, metastatic PC (mPC) still has a poor prognosis mainly due to the development of therapy resistance. In this context, the use of immunotherapy alone or in combination with other drugs has been explored in recent years. However, T-cell directed immune checkpoint inhibitors (ICIs) have shown limited activity with inconclusive results in mPC patients, most likely due to the highly immunosuppressive PC tumor microenvironment (TME). In this scenario, targeting macrophages, a highly abundant immunosuppressive cell type in the TME, could offer a new therapeutic strategy to improve immunotherapy efficacy. In this review, we summarize the growing field of macrophage-directed immunotherapies and discuss how these could be applied in the treatment of mPC, focusing on their combination with ICIs.

## 1. The Therapeutic Landscape of Metastatic Prostate Cancer

Prostate cancer (PC) is the most common malignancy in men and was the fifth cause of cancer death in males worldwide in 2020 [1]. About 80–90% of PC patients are diagnosed at localized or locally advanced stages wherein local treatments, such as surgery or radiotherapy, possibly in combination with androgen deprivation therapy (ADT), can be curative in a high percentage of patients [2]. For PC patients who relapse after local therapy and those with metastatic PC (mPC), ADT is the standard therapy. However, almost all mPC patients eventually progress to incurable metastatic castration-resistant PC (mCRPC), defined as radiographic progression and/or a rise in prostate-specific antigen (PSA), regardless of having a castrate level of testosterone (<50 ng/mL) [3,4]. For decades, therapeutic options for patients with mCRPC have been limited. Since 2004, Docetaxel has been used in the first line chemotherapy-based treatment for these patients [5]. However, over the last ten years the therapeutic landscape of mCRPC has changed dramatically [6] thanks to the discovery and subsequent approval of the second-generation taxane cabazitaxel [7], new androgen receptor signaling inhibitors (ARSIs) such as abiraterone [8,9] and enzalutamide [10], the alpha-emitter radium-223 for patients with symptomatic bone metastasis [11], poly (ADP-ribose) polymerase inhibitors (PARPi) for patients with alterations in DNA damage repair (DDR) genes [12], PSMA radioligands [13], as well as platinum-based treatments recommended for patients with aggressive variants of PC (AVPC) who have progressed on docetaxel treatment [14,15]. Moreover, it is important to note that several of these drugs, especially docetaxel and ARSIs, have been tested and approved in combination with ADT in earlier disease settings, including metastatic castration sensitive prostate cancer (mHSPC) [16,17,18]. However, despite these new therapeutic options, mCRPC still has a poor prognosis, with a median overall survival (OS) of approximately three years, mainly due to disease heterogeneity and the development of therapy resistance. Thus, to overcome these challenges, biomarker-based precision medicine approaches guiding the sequence of systemic therapy as well as new therapeutic strategies are needed. In this setting, an increasing effort has been made in recent years to incorporate immunotherapy in the treatment of mPC, however, with no successful results [19]. In this review we will discuss the current status of immunotherapy in the mPC therapeutic scenario as well as the mechanisms involved in immunotherapy resistance, focusing on PC immunosuppressive tumor microenvironment (TME) and specifically on the role of tumor-associated macrophages (TAMs). Finally, we further discuss the therapeutic potential of targeting TAMs to improve current immunotherapies in mPC.

## 2. Immunotherapy in Metastatic Prostate Cancer Treatment: Current Status and Mechanisms of Resistance

It is well known that immunotherapy, and particularly immune checkpoint inhibitors (ICIs) targeting the T-cell receptor–ligand interaction, such as cytotoxic lymphocyte antigen-4 (CTLA-4), programmed cell death protein 1 (PD-1), or programmed death ligand 1 (PD-L1) [20], have made major advances in the last decade and are widely used in clinical practice to treat urological tumors [21], such as renal cell carcinoma [22] and urothelial cancer [23]. However, in mPC the results have to date been quite modest, except for a small subgroup of mCRPC patients (3–5%) that present a microsatellite instability (MSI) and mismatch repair-deficient (dMMR) phenotype in which exceptional responses to the anti-PD-1 pembrolizumab have been reported [24,25,26]. High-tumoral mutation burden (TMB) is also commonly associated with better clinical outcome to ICIs [27,28]; however, only a small subset (3–8.3%) of mPC tumors have a high TMB, representing a significant obstacle for immunotherapy efficacy [29].

Paradoxically, PC was one of the first diseases for which immunotherapy was approved. Sipuleucel-T, an autologous antigen-presenting cell (APC)-based immunotherapy, was—on the basis of a modest benefit in OS—approved by the Food and Drug Administration (FDA) for asymptomatic and minimally symptomatic men with mCRPC [30]. However, Sipuleucel-T has limited acceptance by the medical community and is not approved by the European Medicines Agency (EMA) due to the complex and costly production of this cellular immunotherapy combined with its limited benefit and the appearance of ICIs in the clinical setting. Nevertheless, as previously stated, several clinical trials have evaluated the efficacy of ICIs in mPC patients who were not selected based on predictive molecular biomarkers, either as single agents or in combination with other checkpoint inhibitors or with other therapies with limited overall activity and inconclusive results [19,31]. Thus, an important current challenge is to improve the efficiency of ICIs in the mPC therapeutic setting. In fact, at present, a large number of immunotherapy-based clinical trials are ongoing in PC, many of which are testing combination therapies or new ICIs [19].

Boosting the immune system may be a good strategy to treat mPC. Nonetheless, it is essential to understand the underlying mechanisms explaining the lack of ICIs efficacy in mPC in order to identify predictive biomarkers as well as new therapeutic targets that could allow us to improve immunotherapy outcome. In this context, several potential mechanisms of PC resistance to ICI immunotherapy have been proposed. In contrast to other tumors such as melanoma or lung cancer, PC has traditionally been recognized as an immunologically “cold” tumor with low levels of tumor-associated antigens and neoantigens, which represent an important mechanism of resistance to ICIs [32]. The expression of the major histocompatibility complex (MHC) class I, a molecule that presents antigenic protein fragments to cytotoxic T cells, is also decreased in PC and consequently limits the proliferation and activation of these cells [33].

Additionally, the PC immunosuppressive TME has also been proposed as a major barrier for immunotherapy efficacy due to the high number of immunosuppressive T cells and the low TME permeability preventing CD8+ cytotoxic T cells and natural killer (NK) infiltration [34].

Taken together, the evidence suggests that mPC cells can escape T-cell recognition through several mechanisms. Additionally, the role of TAMs in immune evasion may be important for treatment resistance, as summarized in the following sections.

## 3. Tumor-Associated Macrophages in PC

Macrophages are versatile cells of the myeloid hematopoietic system that have a wide range of functions, including tissue development, homeostasis, and innate immune responses. In adult tissues, in the absence of inflammation, resident macrophages have an embryonic origin and can persist and replenish locally throughout the adult life [35]. These cells are key to maintaining homeostasis. In the setting of inflammation, infection, or any imbalance requiring increased macrophage activity, blood monocytes raised in the bone marrow can flow out through the endothelium, reach the tissue, and differentiate according to the input of the microenvironment to replenish resident macrophages [35,36,37].

Their differential origin and their elevated plasticity can explain why, although they have many features in common, macrophages are heterogeneous in terms of gene and microRNA expression signatures, epigenetic modifications, surface receptor expression patterns, secretory profiles, and functional properties [38,39]. For example, macrophages can be actively proinflammatory or highly immunosuppressive, depending on environmental cues and molecular mediators [37,40,41]. Prototypic proinflammatory macrophages (often defined as classical or M1 macrophages) can support pathogen and tumor cell killing, are induced by Th1 inflammatory cytokines and/or microbial factors (e.g., IFN/LPS), and can be identified by their surface expression of MHC-II and CD80, the secretion of IL-6 and TNF-α, and an increased expression of iNOS [37,42]. In contrast, immunosuppressive, pro-resolving macrophages (often defined as alternative or M2 macrophages) support the effector functions of Th2 thymocytes and can aid in the later stages of the repair process. They are induced by exposure to Th2 cytokines like IL-4 and IL-13, anti-inflammatory cytokines IL-10 and TGF-β, glucocorticoids or tumor microenvironmental factors, and can be defined by CD163 and CD206 surface expression and increased production of IL-10 or Arg1 [37,42]. However, since the plasticity of macrophages allows the coexistence of intermediate populations, their identification and classification are often challenging, as has been extensively reviewed [37,39,41,42].

Within the tumor, macrophages constitute between 30 and 50% of infiltrating immune cells, therefore, they represent an interesting target for immunotherapy [43]. Moreover, in vivo studies have demonstrated that macrophages mediate both chemo- and immunotherapy resistance through the secretion of soluble factors and the mediation of matrix deposition and remodeling that induce pro-survival and/or anti-apoptotic programs in the malignant cells and the TME [44,45,46].

Specifically, in PC, several studies have reported that TAMs infiltration into the TME supports PC cell proliferation and migration and is associated with disease progression and metastasis after therapy with ARSIs [47,48]. Indeed, TAMs infiltration is often correlated with poor OS of mPC patients [49,50]. Of note, inhibition of androgen receptor (AR) signaling has been correlated with an increased expression of CCL2 along with CCL2-CCR2 axis activation in PC, thereby enhancing metastasis through macrophage recruitment [51,52]. Likewise, Huang and colleagues demonstrated that TAM-secreted CCL5 could promote the migration, invasion and epithelial–mesenchymal transition (EMT) of PC cells as well as the self-renewal of PC stem cells by activation of β-catenin/STAT3 signaling [53].

In addition, PC tumors secrete high levels of growth factors, cytokines and chemokines, including TGF-β, IL10 and CXCL2, which help recruit several immunosuppressive cells (myeloid-derived suppressor cells [MSDC] and regulatory T [Treg] cells) and pro-tumorigenic TAMs, both in the TME and in peripheral blood [54,55], thereby promoting tumor tolerance and evasion by suppressing the proinflammatory type 1 CD4+helper T (Th1) and CD8+ cytotoxic T cells [56]. The key role of cancer-associated fibroblasts (CAFs), the predominant cell type in the TME, in PC tumorigenesis and therapy resistance has also been well described [57]. CAFs are active factors for monocyte recruitment toward tumor cells by stromal-derived growth factor-1 (SDF-1) and CCL2 secretion, promoting their trans-differentiation into the M2 macrophage phenotype [58]. It has similarly been suggested that CAFs are also able to induce the trans-differentiation of proinflammatory and anti-tumorigenic macrophages to anti-inflammatory and pro-tumor TAMs [59].

On the other hand, a loss of PTEN, a negative regulator of the PI3K/AKT/mTOR pathway, is present in approximately 60% of mCRPC patients [60] which is related to worse prognosis, treatment resistance, tumor grade, tumor stage, and risk of recurrence [3,61]. Interestingly, TAMs expressing CXCR2 can infiltrate PTEN-null prostate tumors; CXCL2 activation of CXCR2 can direct macrophages towards an anti-inflammatory phenotype [62].

It has been shown that the infiltration of tolerogenic TAMs in the TME promotes a pro-tumorigenic and immunosuppressive effect due to the secretion of high levels of growth factors, cytokines and chemokines, such as TGF-β, arginase 1 (ARG1), IL10 and CCL20, contributing to a poor infiltration of cytotoxic T lymphocytes and high recruitment of immune suppressive Foxp3+ Tregs cells [63,64,65,66,67]. In fact, a positive feedback loop between TAMs and Tregs enhances their immunosuppressive effects in the TME, as Tregs can enhance the immunosuppressive properties of TAMs and vice versa [68].

Overall, these data indicate that TAMs play a dual role as “tumor promoters” and “immune suppressors”: they can promote tumor initiation and metastasis and act as central drivers of the immunosuppressive TME. Hence, targeting TAMs could represent a potential therapeutic strategy against mPC.

## 4. Involvement of TAMs in ICIs Efficacy

PD-L1 and PD-1 are also expressed in TAMs [69,70,71] promoting immune suppression and escape. ICIs were not originally meant to target macrophages directly, but several studies suggest that macrophages also contribute substantially to the final outcome of these strategies. TAMs express PD-1, with a higher expression in more advanced stages of primary human cancers [69]. PD-1+ TAMs showed a reduced degree of phagocytosis of *S. aureus* bioparticles and tumor cells compared to PD-1− TAMs. Interestingly, the decreased phagocytosis activity of PD-1+ TAMs could be rescued by PD-1/ PD-L1 blockade, which led to a direct decrease in tumor burden [69]. Furthermore, anti-PD-1 or PD-L1 immune checkpoint blockade induced an M1 macrophage polarization [72,73]. A common feature associated with anti-CTLA-4–mediated tumor rejection is an increase in the ratio of T effector to T regulatory cells within the tumor [74,75,76]. Simpson et al. revealed that reduction of Treg cells provoked by anti-CTLA-4 antibodies depend on FcγRIV-expressing macrophage-mediated cell depletion [77]. Therefore, these studies suggest that immune checkpoint inhibitors targeting CTLA-4 or PD-1/PD-L1, in addition to T cells, may also modulate TAM activity in such a way that contributes to antitumor efficacy. In the following sections we will summarize and discuss the current strategies targeting TAMs that are under preclinical and/or clinical investigation.

## 5. Targeting Macrophages in Cancer Therapy and Its Application in PC

Macrophages have emerged as a promising target to directly reduce tumor progression and furthermore overcome immunotherapy resistance. Macrophages can be targeted by different approaches. Historically, the most common approach has been to reduce the number of macrophages at the tumor site, either by affecting their migration capacity and thereby their recruitment to the tumor site, or by inducing macrophage death or depletion. Moreover, recent studies have put their effort on reprogramming macrophages at the tumor site instead of depletion in order to reduce adverse effects. Additional novel strategies include macrophage modulation as adjuvant in vaccine therapies or adoptive macrophage cell therapy.

Of all the treatment modalities presented herein, none have been approved so far for their clinical use in PC. However, several are currently being studied in the context of this malignancy and have reached phase I-IV clinical trials (Table 1).

In this review we provide an overview of the most relevant strategies for targeting TAMs, as summarized in Figure 1. In addition to those strategies that were designed to directly target TAMs, we have also included those developed for other purposes that display modulatory effects on various immune cells, including TAMs.

### 5.1. Therapies Affecting TAM Precursor Recruitment

**CSF-1R.** Colony stimulating factor 1 receptor (CSF-1R) is a transmembrane tyrosine kinase receptor whose expression is restricted to myeloid cells, that dimerizes upon binding of its two ligands, CSF-1 or IL-34 [78]. Dimerization induces a phosphorylation cascade of several macrophage-related signaling pathways that are involved in the differentiation, dissemination, survival, and migration of myeloid cells, including TAMs [79,80]. In humans, CSF-1 increased expression associates with poor prognosis in several cancers, such as gastric cancer, breast cancer, and leiomyosarcoma [81,82,83]. In experimental mouse models of lung adenocarcinoma, targeting CSF-1R with the drug PLX3397 (pexidartinib), which also inhibits two other tyrosine kinase receptors, KIT and FLT3 [84], decreased tumor burden by modification of TAM distribution [85]. Similarly, CSF-1R inhibition with PLX5622 in a mouse model of medulloblastoma, reduced TAMs (IBA1+ microglial cells) around 60%, which reduced tumor growth and prolonged mouse survival [86]. In the clinical settings, inhibiting CSF-1R with PLX3397 showed encouraging preliminary results in a phase I trial, which examined the safety and pharmacokinetics of the agent in patients with advanced, incurable, solid tumors (NCT01004861). Responses were experienced within the first 4 months of treatment and the median duration of response surpassed 8 months. Best responders were those patients with tenosynovial giant cell tumor (TGCTs), a rare tumor type with a central role for CSF-1 in its pathogenesis [87]. Subsequently, in a human phase III trial in TGCTs patients, the administration of PLX3397 significantly reduced the tumor size with a 39% overall tumor response, compared to no tumor response in patients treated with placebo (ENLIVEN Study, NCT02371369) [88].

Regarding PC, androgen blockade therapy and radiotherapy were found to increase myeloid derived suppressor cell (MDSCs) systemically and CSF-1 expression by tumor cells. Subsequently, combination of CSF-1R inhibition with androgen blockade therapy or irradiation reduced tumor progression in subcutaneous PC mouse models [89,90]. At a clinical level, a phase I clinical trial enrolled 36 participants with advanced breast or PC who failed to respond to other treatment modalities. Patients were treated with mAb LY3022855. On the 8th day post first dose administration, 22 participants with metastatic breast cancer and 12 with mCRPC showed increased circulating CSF-1 levels and decreased proinflammatory monocytes. Irrespective of some common adverse events, the treatment was well tolerated and showed evidence of immune modulation (NCT02265536) [91]. Another phase I clinical trial, studying the effects of investigational drug JNJ-40346527, a selective inhibitor of CSF-1R, on patients with high-risk PC that are resectable and showing no signs of local and distant metastasis, is currently active (NCT03177460).

CSF-1R inhibitors have also proved their efficacy in combination with chemotherapy. For example, the CSF-1R inhibitor PXLX3397, in combination with BRAF inhibitor PLX4720, reduced primary and metastatic BRAF-mutated melanoma in a mouse model [92]. Interestingly, in this study, a reduced infiltration of macrophages was observed. Strikingly, this combined inhibition increased PD-1 and PD-L1 expression on CD11b+ cells, making these mice more sensitive to PD-1/PD-L1 inhibitory therapy. Similar results were observed in a mouse pancreatic cancer model, where blockade of CSF-1R reduced TAMs in the TME and reprogrammed these TAMs to increase antigen presentation and T-cell activation, ultimately increasing sensitivity to PD-1/PD-L1 and CTLA-4 immunotherapies [93]. In this context, multiple clinical trials are trying to determine the safety and efficacy of combination therapy between CSF-1R inhibitors with PD-1/PD-L1 or CTLA-4 blockade in advanced solid tumors (NCT02452424, NCT02777710, NCT02829723, NCT02718911, NCT03238027, NCT02880371). Enrollment for the study of the combination of PLX3397 and pembrolizumab (anti-PD-1) to treat advanced melanoma and other solid tumors was terminated early for insufficient evidence of clinical efficacy (NCT02452424). Likewise, using mAb LY3022855 combined with durvalumab (anti-PD-L1) or tremelimumab (anti-CTLA-4) in patients with advanced non-small cell lung cancer (NSCLC) or ovarian cancer (OC) had limited clinical activity, but the treatment was well tolerated (NCT02718911) [94].

**CXCR2**. C-X-C motif chemokine receptor 2 (CXCR2), also known as IL-8RB, is one of the receptors for IL-8, which is mainly expressed in granulocytes and macrophage progenitors [95]. Through its interaction with ligands, CXCR2 mediates a powerful chemotaxis that is associated to tumor progression, angiogenesis, invasion, metastasis and chemoresistance [96,97,98]. Thus, CXCR2 is considered a marker for poor prognosis in many tumor types and, in fact, CXCR2 blockade has been reported to re-educate TAMs and inhibit tumor growth in mouse models of pancreatic ductal adenocarcinoma (PDAC) [62]. Moreover, preclinical studies have highlighted that CXCR2 inhibition not only prevents metastasis formation in PDAC, but also increases the efficacy of anti-PD-1 inhibitors by increasing T-cell infiltration [99]. Similarly, in a lung cancer mouse model CXCR2 inhibition decreased tumor-associated neutrophils, increased anti-tumor T cell activity through enhanced CD8+ T cell activation and increased the therapeutic effect of cisplatin treatment [100]. Thereby, the newly discovered drug SX-682, inhibitor of the CXCR1/2 chemokine receptors, is currently in a phase I clinical trial to determine if it is an effective treatment for metastatic melanoma patients in combination with pembrolizumab (anti-PD-1) (NCT03161431) and whether it could be beneficial in combination with nivolumab (anti-PD-1) as a maintenance therapy in subjects with metastatic PDAC (NCT04477343) or metastatic colorectal cancer (NCT04599140). Likewise, a study on patients diagnosed with NSCLC, CRPC and microsatellite stable colorectal cancer, using navarixin (MK-7123), a CXCR1/2 antagonist in combination with pembrolizumab (anti-PD-1), has completed its phase II trial (NCT03473925). Another multi-centric proof of concept, phase I/II clinical trial, with CXCR2 antagonist AZD5069 in combination with the AR antagonist enzalutamide involving individuals with mCRPC is currently ongoing (NCT03177187).

**CCL2-CCR2.** The monocyte chemoattractant protein-1 (MCP-1), also known as CCL2, is a cytokine secreted by various cell types that drives the migration of myeloid and lymphoid cells after exposure to an inflammatory stimulus [101]. CCL2 binds the cognate receptor CCR2, and together this signaling pair has been shown to have multiple roles. In the context of cancer, it recruits immune cells to the tumor site and induces tumor cell proliferation, angiogenesis and metastasis [102]. Likewise, increased expression of CCL2 is associated with the accumulation of TAMs in esophageal cancer, both being predictors of poor prognosis [103]. Further, CCL2 affects TAM activity in many ways that affect tumor behavior. For instance, CCL2 expression by TAMs promoted the acquisition of an invasive phenotype in breast cancer [104].

CCL2 has been suggested as a potential biomarker for several types of cancer, including prostate [105,106,107,108]. Given the critical roles of the CCL2-CCR2 signaling axis in tumorigenesis, a series of clinical trials targeting this axis have been carried out in several cancers [102,107]. Results so far, however, have not been promising. For example, since targeting CCL2 with the mAb carlumab did not have a significant effect, it was suggested that anti-CCR2 may be a compensatory approach. In pancreatic cancer, CCR2 antagonists—either alone or in combination with chemotherapy—were able to control local tumors and were well tolerated in patients [107].

Only one clinical trial has been conducted with PC patients, a phase II, open-label, multicentric trial involving participants with mCRPC. Similar to previous assays in other solid tumors, there were no complete or partial responses after carlumab administration, so the study did not progress (NCT00992186). It has been proposed that the weak affinity of carlumab with CCL2 and the inadequate clearance of the circulating CCL2-complex may be among the reasons for the unsatisfactory therapeutic effects [107].

In several cancer models, the CCL2–CCR2 signaling axis could induce tumor immune evasion through PD-1 signaling, thus promoting TAM-mediated immune evasion [103,107]. In light of these results, it might be promising to move to combination regimen with anti-PD-1 into a clinical trial in the near future.

**GM-CSF.** The granulocyte–macrophage colony-stimulating factor (GM-CSF) was originally identified as a colony stimulating factor because of its ability to induce granulocyte and macrophage populations from precursor cells. Multiple studies have demonstrated that GM-CSF is also an immune-modulatory cytokine, capable of affecting not only the phenotype of myeloid lineage cells, but also T-cell activation through various myeloid intermediaries [109]. Besides, several studies suggest that GM-CSF has antitumor activity [110,111,112], which could be due to GM-CSF-induced M1 macrophage polarization and macrophage activation [39,113,114]. Accordingly, several strategies have been developed for GM-CSF-based cancer immunotherapy in clinical practice, including GM-CSF therapies and GM-CSF-based DNA vaccines [115]. GM-CSF has been extensively assayed in PC clinical trials (ClinicalTrials.gov).

In the 1990s, the FDA approved the use of GM-CSF to treat various types of cancer patients with chemotherapy-induced neutropenia and leucopenia, and it was widely used achieving excellent results [116]. Focusing on PC therapy, several studies have been conducted so far with GM-CSF. Remarkably, a phase II study with 125 participants for metastatic hormone refractory PC, showed that the administration of GM-CSF as maintenance therapy after docetaxel + prednisone chemotherapy increased the OS from 14 to 28.4 months when compared to patients receiving only chemotherapy (NCT00488982). Furthermore, there are a few ongoing clinical trials in phase II and III aiming to minimize adverse effects of chemotherapy and increasing OS (NCT04709276 and NCT02961257, respectively).

GM-CSF has also been used as a vaccine adjuvant with success. This is the case of Sipuleucel-T, the first therapeutic vaccine approved by the FDA for the treatment of patients with asymptomatic or minimally symptomatic mCRPC. This autologous mononuclear cell immunotherapy is formulated to stimulate an immune response to PC cells targeting prostate acid phosphatase and induce antigen-specific T cells [117,118]. Combination with GM-CSF prolonged the survival of patients in several clinical trials [118].

Additionally, GM-CSF is being tested as an adjuvant of DNA vaccines. In a phase I clinical trial involving participants with mPC, administration of a DNA vaccine encoding androgen receptor ligand-binding domain with GM-CSF prolonged progression free survival (PFS) (NCT02411786) [119]. Besides, in a phase II clinical trial, non-metastatic prostate cancer patients that were treated with a DNA vaccine encoding prostatic acid phosphatase (pTVG-HP) and GM-CSF did not exhibit an overall increase in the 2 years of metastatic-free survival, but showed considerable effects on the micro metastatic bone disease (NCT01341652) [120]. Moreover, additional clinical trials are recruiting to test the efficacy of GM-CSF with pTVG-HP in combination with the anti-PD-1 nivolumab (NCT03600350) and cabazitaxel plus prednisone (NCT02961257). Finally, phase I/II clinical trials are currently active to evaluate the therapeutic vaccines Proscavax (PSA/IL2/GM-CSF) and UV1 (synthetic peptide) + GM-CSF, but they are not yet recruiting (NCT03579654 and NCT01784913, respectively).

**Tasquinimod.** The quinoline-3-carboxyamide tasquinimod is a small molecule immunotherapy with demonstrated pleiotropic effects on the TME. It binds and inhibits the interactions of the damage-associated molecular pattern receptor S100A9 (S100 Calcium Binding Protein A9), a key cell surface regulator of myeloid function [121,122]. Tasquinimod affects tumor infiltrating myeloid cells rapidly after exposure, leading to a change in phenotype from pro-angiogenic and immunosuppressive TAMs to proinflammatory macrophages [121]. It has shown antitumor, anti-angiogenic and immune-modulatory properties in several murine models of solid tumors, including PC [121,122,123,124]. Tasquinimod is the most studied agent for PC treatment, as it has been evaluated in several clinical trials as a single agent as well as in combination therapy with other systemic agents in mCRPC [123,125,126,127,128].

For example, a phase II clinical trial was conducted on individuals diagnosed with mCRPC who did not exhibit any signs of progression after receiving the first dose of docetaxel therapy. The results demonstrated a median PFS of 31.7 weeks in the treated and 22.7 weeks in the placebo arm, additionally, when used as maintenance therapy, tasquinimod efficiently reduced the risk of radiologic progression-free survival (rPFS) by 40% (NCT01732549) [127]. Similarly, another phase II randomized double-blinded study using tasquinimod (ABR-215050) was conducted on 201 mCRPC individuals. The drug was well tolerated by the patients with a median OS of 34.2 months in the treatment and 27.1 in the placebo group. The exploratory biomarkers correlated with the survival and drug efficacy (NCT00560482) [129,130]. A double-blinded, phase III randomized trial using tasquinimod was conducted on 1245 patients with mCRPC. The results demonstrated a median rPFS of 7.0 months in the treatment arm when compared to 4.4 months in the placebo. However, no significant changes in the OS were observed. The authors of the study suggested that the identification of predictive biomarkers of tasquinimod efficacy may contribute to increasing OS in the future (NCT01234311) [131].

**FAK.** Focal adhesion kinase (FAK) is a non-receptor tyrosine kinase involved in multiple cellular and extracellular processes, namely macrophage adhesion, and is required for the formation of lamellipodia, migration, and infiltration into inflamed areas [132]. It is upregulated at mRNA and protein levels in many advanced stage solid tumors [133], like ovarian serous cystadenocarcinoma, head and neck squamous cell carcinoma, and prostate adenocarcinoma (data from The Cancer Genome Atlas). In the context of cancer, FAK participates in several processes involved in metastasis progression, from the expression of matrix metalloproteinases and ECM remodeling, to focal adhesion formation and turnover [134]. Preclinical and clinical studies have outlined the benefits of FAK inhibition, as it not only affects tumor cells but also other cells from the TME like fibroblasts [135]. Moreover, FAK inhibition reduces MDSCs, TAMs and Tregs infiltration within the tumor in mouse models of squamous cell carcinoma [136] and pancreatic cancer [137]. In both genetic and syngeneic mice models of PDAC, combination of the FAK inhibitor VS-4718 with the chemotherapeutic agent gemcitabine and anti-PD-1 and/or anti-CTLA-4 therapy achieved a maximal response in terms of reducing tumor progression and increasing survival. This was due to the increased numbers of CD8+ cytotoxic lymphocytes that infiltrate the stroma and reach the tumor as well as the reduced numbers of Tregs [137]. Hence, targeting FAK improved response to existing therapies, and it was proposed that it may serve to overcome treatment resistance.

To date, 12 phase II clinical studies with FAK inhibitors have been registered, among them, NCT02004028, in which brief preoperative defactinib exposure in malignant pleural mesothelioma patients was well tolerated, did not alter resectability or mortality compared to prior series, and showed evidence of therapeutic and immunomodulatory effects. Biological correlates of treatment included target inhibition (75% pFAK reduction); tumor immune microenvironment changes: increased naïve (CD45RA+PD-1+CD69+) CD4 and CD8 T cells, reduced myeloid and Treg immuno-suppressive cells, reduced exhausted T cells (PD-1+CD69+), reduced peripheral MDSCs; and histological subtype change (pleomorphic or biphasic to epithelioid) in 13% of cases [138]. Additionally, FAK inhibition in combination with other therapies, such as the PD-1 inhibitor pembrolizumab, is currently being tested in advanced solid cancers (NCT03727880, NCT02758587 NCT02523014 and NCT03287271 NCT04620330). Altogether, the capacity of FAK of modulating the environment within the tumor has potential as anti-tumor treatment and in combination to improve efficacy of both immunotherapy and chemotherapy. However, to our knowledge, FAK inhibition has not been tested on PC patients.

### 5.2. Therapies That Induce Depletion or Affect Macrophage Survival

**Bisphosphonates.** These inorganic compounds, including clodronate and zoledronic acid among others, are structurally identical to bone matrix pyrophosphatases, which is why they are easily metabolized by osteoclasts and inhibit their resorption. They have been used in preclinical bone metastasis models and are already in use for the treatment of hematological and solid malignancies [139]. Bisphosphonates mainly affect osteoclasts, which share lineage with macrophages. However, they also inhibit proliferation, migration and invasion of macrophages, inducing apoptosis [140]. Moreover, bisphosphonates also inhibit cancer cell proliferation, adhesion, and invasion; induce tumor cell apoptosis; block angiogenesis; and interfere with immune surveillance [141]. Large trials have clearly demonstrated the clinical value of different bisphosphonate-based drugs (including the oral drugs ibandronate and clodronate and intravenous agents such as zoledronate and pamidronate), in treatment of the hypercalcaemia of malignancy and the reduction of skeletal-related events (SREs) and symptomatic skeletal events (SSEs) in a range of cancers. Bisphosphonates also remain mainstay of drugs for the treatment of metastatic bone disease, which has recently been reviewed in [142].

Bisphosphonates have long been studied and used as an interventional therapy in PC. Zoledronic acid, the most widely used bisphosphonate, is in advanced trial phases (phase III/IV). While some studies indicate inefficacy of zoledronic acid when used as a treatment modality, alone or as a combination, in advanced stages (NCT00365105) [143], (NCT00079001) [144], (NCT00869206) [145], other analyses support its use in PC as a post-chemo-maintenance therapy (NCT00554918) [146] and to reduce skeletal-related events [147], (NCT00321620) [148]. Several clinical trials using zoledronic acid are currently recruiting volunteers (NCT00268476, NCT03336983 and NCT04549207). Few meta-analysis studies have also been performed to evaluate the effects of bisphosphonates therapy in PC individuals with bone metastasis [149,150,151], and results from 18 clinical trials suggested that bisphosphonates therapy had a high probability of decreasing the skeletal-related events and disease progression [150]. In conclusion, at the clinical level, trials on PC have given inconsistent results, suggesting a need for treatment combination optimization.

**Trabectedin.** This tetrahydroisoquinoline alkaloid alters transcriptional regulation by binding to and inducing DNA damage, affecting slow-dividing cells and cells in the G1 phase of the cell cycle [152]. Trabectedin targets tumor cells, but also monocytes and macrophages within the tumor, which are quiescent and thus poorly sensitive to classical DNA-damaging agents. It induces apoptosis by activation of caspase 8 through TRAIL, a TNF-related apoptosis-inducing ligand [153]. Interestingly, resident tissue macrophages have low expression levels of functional TRAIL receptors, while TAMs from liver, colon and breast carcinoma mainly express the functional TRAIL-R2 [154]. Trabectedin is currently approved by the EMA for the treatment of advanced soft-tissue sarcoma and platinum-sensitive relapsed OC in combination with PEGylated liposomal doxorubicin (PLD) [155] and by the FDA for metastatic or unresectable liposarcoma or leiomyosarcoma following an anthracycline-containing regimen [156]. Although there are currently several clinical trials on trabectedin, most of them aim to improve the response in sarcomas (TARMIC study, NCT02805725; NiTraSarc, NCT03590210), or in combination in recurrent OC (NCT03470805, NCT04887961). This suggests limited future perspectives for trabectedin as a broader anti-cancer therapy.

Regarding PC, clinical trials have also been completed using trabectedin as a treatment modality (NCT00147212, NCT00072670). In these trials, trabectedin showed modest activity in mCRPC. The authors of the studies state that identification of predictive factors of response in PC could contribute to better outcomes [157].

### 5.3. Strategies to Reprogram Macrophage Activity

**CD47-SIRPα.** CD47 is a ubiquitous protein that regulates cytokine production, T-cell activation, and cell migration [158] through its interaction with thrombospondin-1 and signal regulatory protein-α (SIRPα), which are mainly expressed by dendritic cells and macrophages [159]. This interaction results in a “do not eat me” signal that prevents phagocytosis of autologous cells in homeostatic conditions. This mechanism is tightly regulated, and it is mainly activated under proinflammatory conditions [160]. CD47 is overexpressed in a variety of tumors [161,162,163], and is involved in tumor invasion, metastasis, and the inhibition of phagocytosis by interacting with SIRPα-expressing phagocytes [162,163]. In fact, preclinical studies in mouse models have shown that inhibition or blockade of CD47 inhibits tumor growth and enables the phagocytosis and killing of tumor cells by macrophages in several types of tumors, including ovarian, endometrial, liver, and squamous cell lung cancer, respectively [164,165,166,167,168]. Moreover, targeting CD47 in a mouse model of esophageal squamous cell cancer not only enhanced proinflammatory responses and increased infiltration of CD8+ T cells within the tumor, but also increased PD-1 and CTLA-4 expression, thus increasing mouse susceptibility to anti-PD-1 and CTLA-4. Hence, combination of anti-CD47 with anti-PD-1 and CTLA-4 resulted in the best antitumor response [169]. The growing interest in targeting the CD47-SIRPα interaction is reflected by the many strategies being tested at present, which include humanized anti-CD47 mAbs magrolimab (also named Hu5F9-G4), AK117 and AO-176, or anti-CD47/PD-1 or /PD-L1 bifunctional antibodies (HX009 and PF-07257876, respectively) as well as recombinant fusion proteins SIRPα-Fc (TTI-621, and Evorpacept or ALX148) and SIRPα-Fc-CD40L (SL-172154). At present, 46 clinical trials are studying the safety and efficacy of these drugs, either alone or in combination with other immunotherapies and/or chemotherapy. For example, in advanced solid tumors and hematologic malignancies, SIRPα-Fc protein is being tested in combination with the chemotherapy doxorubicin or with anti-PD-1 mAbs (NCT02663518, NCT04996004). In addition, combination of anti-CD47 mAb magrolimab with anti-PD-L1 mAb is under phase I clinical trials for checkpoint-inhibitor-naive OC patients, who progressed within 6 months after platinum chemotherapy (NCT03558139), while combination therapy of magrolimab with anti-EGFR mAb cetuximab is being tested for advanced colorectal cancer (NCT02953782). Taken together, current data suggest that the combination of CD47 blockage with other therapies may be a promising strategy to treat advanced tumors that until now could not benefit from immunotherapy, including mCRPC.

**Adoptive cell therapy.** Application of chimeric antigen receptor T cells (CAR-T), approved by the FDA for acute lymphoblastic leukemia and refractory non-Hodgkin lymphoma, is challenging in solid tumors because of the low infiltration of CAR-T cells into the immunosuppressive TME. Taking macrophages into account may improve this modified T-cell therapy. In addition, it is also possible to modify macrophages. The concept of a “chimeric antigen receptor” in macrophages (CAR-M) was first introduced by Morrissey et al. [170] and later validated by Klichinsky et al. [171]. The first generation of CAR-M combined the scFv of anti-CD19, anti-mesothelin, or anti-HER2 antibodies with a CD3 intracellular domain. This CAR-M displayed strong anti-tumor activity in preclinical models; a single infusion of human CAR-M decreased tumor burden and prolonged OS in two solid tumor xenograft mouse models [171]. Accordingly, two clinical trials with CAR-M are already underway in solid tumors. As part of a phase I, first-in-human, open label study, subjects with HER2-overexpressing solid tumors are being recruited to assess the safety and tolerability, as well as response rate and PFS of CT-0508, an anti-HER2 CAR-M (NCT04660929). Furthermore, one recent prospective observational study aims to collect breast tumor samples in order to develop patients’ derived organoids to test the antitumor activity of newly developed CAR-M (NCT05007379).

## 6. Discussion and Future Perspectives

Macrophages represent the largest fraction of immune cells infiltrating tumors, including PC. Moreover, they play a key role in treatment resistance contributing to immune evasion and promoting an immunosuppressive TME. Consequently, the various pro-tumoral mechanisms of TAMs provide many attractive therapeutic targets for cancer treatment. Macrophage-targeting strategies based on recruitment, survival, depletion, and/or reprogramming have shown successful results in clinical trials in advanced solid tumors, indicating that they could possess great therapeutic potential for mCRPC as well.

Regarding therapies affecting macrophage recruitment, there are treatments that have shown positive results in PC clinical trials, highlighting GM-CSF as a therapeutic vaccine adjuvant and CSF-1R as monotherapy, although trials in other solid tumors have suggested that combination therapy of both strategies could improve results obtained up to now. Conversely, although preclinical results indicated an important role of FAK in PC, to our knowledge, FAK inhibitors have not yet been tested in clinical trials in PC. The cytokine CCN3, secreted by PC cells, induces M2 polarization in RAW264.7 macrophages through the activation of the FAK/Akt/NF-κB pathway, leading to an increase in VEGF expression and promoting angiogenesis [172]. Likewise, high levels of macrophage inhibitory cytokine-1 (MIC-1), a member of the TGF-β superfamily, induced PC cell metastasis through the upregulation of the FAK–RhoA signaling pathway [173]. Overall, these results suggest that FAK inhibition could be a potential strategy in PC. In terms of macrophages survival/depletion approaches, it is worth mentioning that although bisphosphonates have been widely studied with controversial results, they should be taken into consideration as they are used to treat metastasis in the bone, the most frequent metastatic site for PC.

Furthermore, newly emerging therapies reprogramming macrophages are in earlier stages, but the data currently available suggest that a new perspective in macrophage-based therapies has opened. The combination of CD47 blockade with chemotherapy and/or other ICIs may be a promising strategy to treat advanced tumors that currently cannot benefit from immunotherapy, including mPC. In addition, other novel TAM reprograming molecules are currently under preclinical investigations. In this regard, CD24 blockade seems an attractive therapeutic alternative for tumors that only respond weakly to anti-PD-1/PD-L1 immunotherapies [174,175]. CD24 is a surface protein that interacts with Siglec-10, which is expressed on innate immune cells, especially macrophages, inducing an inhibitory anti-phagocytic “do not eat me” signaling cascade that attenuates inflammatory responses and the cytoskeletal remodeling required for macrophage phagocytosis, thus inducing cancer cell protection [176,177]. Blocking the CD24-Siglec-10 interaction resulted in tumor growth reduction and prolonged survival in mice models [178]. Finally, the novel emerged technology CAR-M opens new paradigms for the development of personalized macrophages-based cancer immune medicine.

## 7. Conclusions

In conclusion, macrophages represent a potential strategy worth considering in tumor immunotherapy. Even though immunotherapy has shown modest results in mPC, macrophages have been shown to influence the TME and unlock new strategies to overcome treatment resistance. Hence, more approaches targeting macrophages in combination with other immunotherapies need to be explored in the future.

## Figures and Tables

**Figure 1 cancers-14-00440-f001:**
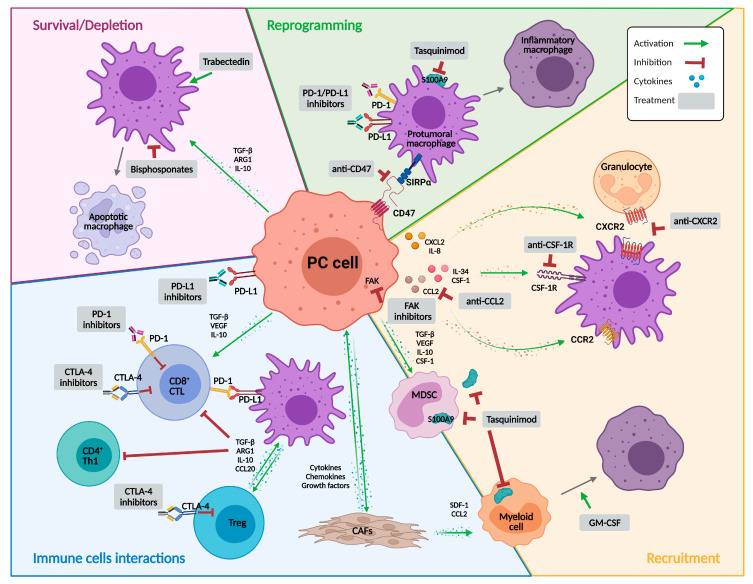
Targeting macrophages as a therapy for prostate cancer. Representative illustration summarizing the immune landscape in the context of prostate cancer and the different potential approaches to target macrophages. The strategies are classified into molecules that mainly affect macrophage recruitment or survival, induce their depletion, or reprogram their activity. Abbreviations: ARG1, arginase 1; CAFs, cancer-associated fibroblasts; CCL2, C-C motif chemokine ligand 2; CCL20, C-C motif chemokine ligand 20; CCR2, C-C motif chemokine receptor 2; CD4+ Th1, type 1 CD4+ helper T cell; CD8+ CTL, CD8+ cytotoxic T lymphocyte; CSF-1, colony stimulating factor 1; CSF-1R, colony stimulating factor 1 receptor; CTLA-4, cytotoxic T lymphocyte-associated protein 4; CXCL2, C-X-C motif chemokine ligand 2; CXCR2, C-X-C motif chemokine receptor 2; FAK, focal adhesion kinase; GM-CSF, granulocyte macrophage colony-stimulating factor; IL, interleukin; MDSC, myeloid derived suppressor cell; PC, prostate cancer; PD-1, programmed death 1; PD-L1, programmed death ligand 1; S100A9, calcium binding protein A9; SDF-1, stromal-derived growth factor-1; SIRPα, signal regulatory protein-α; TGF-β, transforming growth factor β; Treg, regulatory T cells; VEGF, vascular endothelial growth factor. Created with BioRender.com, accessed on 22 December 2022.

**Table 1 cancers-14-00440-t001:** Macrophage-related therapies under clinical trials currently active in prostate cancer.

Intervention ^1^	Phase	NCT ^2^ Number	Indication
CSF-1R inhibitor (JNJ-40346527)	I	NCT03177460	High-risk localized prostate cancer
Enzalutamide + CXCR2 inhibitor (AZD5069)	I/II	NCT03177187	mCRPC
UV1 synthetic peptide vaccine + GM-CSF	I/IIa	NCT01784913	mPC
Carboplatin + GM-CSF Cabazitaxel + GM-CSF	II	NCT04709276	Metastatic prostate neuroendocrine carcinoma and mPC
DNA vaccine pTVG-HP + nivolumab + GM-CSF	II	NCT03600350	Non-metastatic, non-castrate prostate cancer
ProscaVax (PSA/IL-2/GM-CSF)	II	NCT03579654	Localized prostate cancer
Cabazitaxel + prednisone + GM-CSF	III	NCT02961257	mCRPC previously treated with a docetaxel-containing regimen
Sipuleucel-T (APCs loaded with the fusion protein PAP linked to GM-CSF)	III	NCT03686683	Non-metastatic prostate cancer
Enzalutamide and luteinizing hormone-releasing hormone analogue (LHRH-A) + zoledronic acid	II	NCT03336983	mPC
ADT + zoledronic acid ADT + zoledronic acid + docetaxel + prednisolone ADT + zoledronic acid + celecoxib	II/III	NCT00268476	Hormone-naïve prostate cancer
Zoledronic acid	IV	NCT04549207	Bone metastases from breast cancer and CRPC
Pamidronate			
CAR-M (CT-0508)	I	NCT04660929	HER2 overexpressing solid tumors, including PC

^1^ Treatments affecting macrophages. ^2^ National Clinical Trial.

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
