# Peer review of "Macrophages as a Therapeutic Target in Metastatic Prostate Cancer: A Way to Overcome Immunotherapy Resistance?"

_cancers, 2022, doi:10.3390/cancers14020440_

Round 1
Reviewer 1 Report
Please mark in bold letters in chapter 5.1 at the beginning the therapies your are talking about. FAK, GM-SF etc...
Reviewer 2 Report
1. Macrophages have the several subtypes and surface markers. Please add the paragraph regarding the subtypes and the surface markers with their function. Table summarizing these is also desirable.
2. Discuss the function of macrophage in tumorgenesis and localized prostaet cancer.
Reviewer 3 Report
Major comments:
- The aim of the review is to summarize macrophage-directed immunotherapies (Line 35). However, not all the therapies presented in the review are directly targeting macrophages, thus it was confusing to read initially. Section 5 starts off by talking about using therapies that target macrophage and this gives the impression that the therapies to be discussed next are developed specifically for targeting macrophages. Instead, the first target to be discussed in the subsequent section (5.1) was FAK of which FAK inhibition reduces infiltration of various immune cells besides TAMs. In the whole review, the only therapy that seems to be directly targeting macrophages are: GM-CSF, CXCR2, CCL2-CCR2, CD47-SIRPa. Perhaps these should be presented first, before discussing the indirect therapies. The indirect therapies should be described as being developed for other purposes but displayed immuno-modulatory effects on various immune cells that are not limited to macrophages.
- The sub-section headings 5.1 & 5.2 mentions both macrophage & monocytes, but the title of the review is only of macrophages. Is this review meant to be about both macrophages and monocytes?
- An introduction and a lineage diagram defining what macrophages are and how they are derived and their cytokines would be useful for readers without much immune background. E.g. the difference between macrophages, monocytes, APCs, dendritice cells; significance of cytokines related to macrophages, such as GM-CSF as this featured in Table 1.
- It is not very clear which cells in Figure 1 are the macrophages as not all the cells are labelled. Are all the purple cells macrophages? Line 175: "PD-1+ TAM phagocytosis can be rescued...". This sentence needs more clarification as it is not very clear and perhaps missing a word. What is the significance of PD-1+ phagocytosis? Is the phagaocytosis referring to phagocytosis of cancer cells by the TAM? "rescued by PD-1/PD-L1 blockade" seems to imply PD1 is doing something to the phagocytosis but the information is not clear.
- Statement in lines 182- 183: "..TAMs also play a prominent role in immunotherapy amd contribute to anti-tumor efficacy". This statement is too strong as the evidence presented in this section is insufficient to convincingly indicate so. There are too many immune cells affected, it is not clear how much of a role the macrophages play.
- Title and headings of Table 1 can be improved. The title of Table 1 gives the wrong impression that the therapies undergoing clinical trials are primarily designed to target macrophages, which is not the case. The term "macrophage-related therapy" would be more appropriate. The items displayed in the first column ("Target") are not the targets of the therapy being tested. For example, bisphosphonates is a class of compounds, not the target of the therapy. Instead of having separate columns called "Target", "Drug" and "Treatment combinations", the authors should combine them into a single column called "Treatment" or "Intervention", and then highlight in bold treatments that can influence macrophages, noting it in the title or as footnote under the table. Using the first row as an example, the "Treatment" would be "UV1 synthetic peptide vaccine in combination with GM-CSF".
Another column summarising how the main mechanism of the macrophage-related therapy and how it influences the macrophages would be useful too, e.g. bisphophonates - inhibits osteoclast-mediated bone resorption by , <how does it influence macrophages?
Minor comments:
1. Line 44: "second cause of cancer death". Worldwide and which year? Authors should state where and when according to their citation as the statistics differ every year.
2. Line 55: "thanks to the approval". This is not the appropriate phrase to use here. The main reason that therapeutic landscape has improved is because of the development (or discovery) of new effective treatments which then subsequently result in their approval.
3. Sentence in line 78-82 needs to be rephrased as reading it gives the impressio that PD-L1 is a T-cell receptor which is not the case.
4. The sentence "Sipuleucel-T, and autologous...is on the basis of a modest OS benefit".. (Line 91-94) coud be rewritten as it gramatically confusing.
5. Define EMA (Line 95)
6. Lines 19 & 98: "unselected". What is this referring to? Unselected for what?
7. Citation needed for statement in line 123-124.
8. Is simple summary meant to be layman summary? It yes, it should be simplified further.
Round 2
Reviewer 3 Report
The manuscript is significantly improved after revision. I recommend acceptance for publication.